# Probabilistic Maintenance Cost Analysis for Aged Multi-Family Housing

**Moonsun Park [1], Nahyun Kwon [1] , Joosung Lee [1], Sanghyo Lee [2] and Yonghan Ahn [1],***

[1] Department of Architectural Engineering, Hanyang University, Ansan 15588, Korea; cemmoon@empas.com (M.P.); envy978@hanmail.net (N.K.); js4ever@hanyang.ac.kr (J.L.)

[2] Division of Architecture and Civil Engineering, Kangwon National University, Samcheok 25913, Korea; leesh0903@kangwon.ac.kr

* Correspondence: yhahn@hanyang.ac.kr; Tel.: +82-31-436-8182

**Abstract:** To realize sustainable construction, planning for future maintenance costs is essential. In the case of multi-family housing, various maintenance issues can be expected to appear starting 10 years after completion. Therefore, preventive maintenance must be implemented in a systematic manner to cope with the problems caused by the natural aging of multi-family dwellings and to maintain a sustainable level of quality for the properties. In this study, maintenance costs were investigated for 224 multi-family housing units aged 20 years or older in Seoul, South Korea. Using Monte Carlo simulation in conjunction with expert interviews, a probabilistic maintenance cost analysis was conducted to analyze and estimate the variability in maintenance costs. The findings of the study propose that the use of probabilistic maintenance cost analysis can be developed into a useful planning tool for determining reasonable future maintenance costs in sustainable construction.

**Keywords:** multi-family housing; sustainable construction; maintenance costs; probabilistic cost analysis; Monte Carlo simulation

## 1. Introduction

The number of aged multi-family housing units is gradually increasing worldwide. In the U.S., the proportion of these units aged 30 years or older increased from 47% in 1994 to 65% in 2016, representing half of the total stock [1–3]. In Japan, the number of multi-family buildings aged 40 years or older is expected to increase from 430,000 in 2014 to 2,770,000 by 2034 [4]. In South Korea, the number of multi-family buildings aged 30 years or older is also rapidly increasing as a result of the urban developments of the 1980s [2]. As of 2016, 37.4% of the 1,641,383 apartments in Seoul, South Korea (613,502 units) were aged 20 years or older. The trend analysis for the last five years shows that the number of apartments in Seoul will reach 1,790,000 as of 2021, assuming an increase of 30,000 units per year. Among them, the number of apartments aged 20 years or older is expected to reach 950,000, representing more than half of the total number of units [5,6].

Multi-family apartment buildings are subject to issues, such as concrete neutralization, steel reinforcement corrosion, and the failure of various facilities, typically starting from 10 years after completion [7–9]. Preventive maintenance must be systematically implemented to cope with such problems caused by the aging of apartment buildings and to maintain a sustainable standard of quality [10]. For most multi-family housing units, however, planned repairs are not performed due to the non-accumulation of maintenance data, lack of experience in professional process maintenance, and lack of interest of the residents [11]. Therefore, maintenance costs increase as a result of safety accidents caused by functional damage to the main parts of the building, repair of main members, and replacement of consumables [5,12].

Thus far, various efforts have been made to address the problems raised in relation to aged multi-family housing units [13–17]. In the field of construction, studies have been carried out to analyze maintenance costs using various research methodologies in order to enhance the sustainable use of facilities [2,3]. However, due to limitations in terms of concreteness and quality of cost analysis, they have not effectively supported efforts to prevent the deterioration of multi-family housing units through planned repairs. Therefore, a study from an empirical perspective is required to conduct a probabilistic analysis of the maintenance costs related to aged multi-family housing from both planning and economic perspectives.

This study seeks to address this need by analyzing the maintenance cost data of 224 properties to address the problems that prohibit preventive maintenance in Seoul. As buildings typically start to require replacement and repairs 10 years after completion due to the wear and aging of building structures, facilities, and interior/exterior materials, the scope of research was limited to multi-family housing units aged 20 years or older. Section 2 provides an overview of prior studies. Sections 3 and 4 detail the data used and elaborate on the use of Monte Carlo simulation for the probabilistic cost analysis. Section 5 discusses the results of the analysis, and Section 6 provides concluding remarks.

## 2. Literature Review

The maintenance of a building is important for providing users with sustainable usability [2,3,7,18,19]. Currently, the number of aged facilities is on the rise worldwide, making it a major issue for most countries [2,3,7,20,21]. For buildings, the function and performance of facilities begin to deteriorate at a certain point after completion [3,7]. Therefore, periodic and continuous management is required to maintain the functionality and usability of the building. Moreover, an appropriate budget is required for proper planned maintenance, and the development of such a budget requires an in-depth analysis of maintenance costs [22].

In the field of construction, many studies related to maintenance have focused on the timing of maintenance, problems with maintenance, and analysis of maintenance costs [2,3]. To secure the quality and safety of the buildings for the residents, reasonable maintenance costs must be more accurately estimated through the analysis of the past maintenance cost data. In this study, maintenance costs were analyzed using a probabilistic method to emphasize the necessity of managing such costs for aged multi-family housing units.

Kang et al. [23] established a classification system for each multi-family housing unit, and through this system, they investigated cases, examined construction costs using the present value method, and analyzed the difference between the long-term repair reserve deposited each year and the actual repair expenses incurred. Kim [24] analyzed the types of expenditure for repair and maintenance, as well as repair items, by investigating and collecting the maintenance costs inputted into public rental housing cases, and also analyzed the effect of repair and maintenance costs on the rental housing project. Several works [2,3,7,25] have utilized a risk matrix-based method by investigating the maintenance history of multi-family housing units and analyzing the frequency of occurrences and costs of maintenance activities using the Loss-Distribution Approach method. Lee [26] proposed a method of improvement for the establishment of long-term repair plans by investigating the four-year repair cases in multi-family housing units in Seoul and analyzing the frequency of repair occurrences based on the long-term repair plan establishment criteria presented by the Housing Act of South Korea. Other studies [2,3] analyzed the life cycle for mechanical, electrical, and plumbing elements among the components of residential buildings by investigating rental-housing cases in South Korea using a probabilistic approach. Researchers at Harvard University [27] proposed an operating cost prediction model for public housing through regression analysis by classifying the number of households, number of elapsed years, household size, building type, area, and financial support ratio. Goodman [28] analyzed the decisive factors of the operating costs for rental housing by classifying the quality of housing, number of elapsed years, number of households, and area according to the operating and maintenance costs. Muyingo [29] investigated and analyzed the maintenance costs of public rental

housing in Sweden and derived the maintenance costs per unit area of private and public rental through a survey with experts.

As noted previously, previous studies have focused on the maintenance cost analysis method with respect to long-term repair timing and risk for aged multi-family housing units [2,3,7]. While some studies were conducted on repair cycle analysis in accordance with the classification suggested by the long-term repair plan establishment criteria, studies on maintenance cost analysis are still not sufficient. In particular, the lack of analysis using a probabilistic method has resulted in inherent weaknesses in the budget development process for planned maintenance.

Therefore, this study aims to investigate and analyze the maintenance cost of multi-family apartment buildings aged over 20 years in Seoul. The method proposed in this study can be used as a reference for planning maintenance costs as it estimates and analyzes the probabilistic cost fluctuation range by reflecting the opinions of experts and existing maintenance cost data.

## 3. Research Methodology

In the field of construction, there are numerous stakeholders and a series of overlapping processes [30]. Therefore, it is not possible to accurately predict the results of a series of tasks, and probability models are used mainly as tools for analytical research [31–33]. The probability model method is an analytical method, and in many cases, it is impossible to find solutions. In such cases, the Monte Carlo simulation method, which includes simulation analysis by repeatedly using and generating a series of random numbers, is the most influential method for finding accurate results [34].

In general, if multiple data can be used, normal distribution or a log-normal distribution can be determined through statistical analysis. However, if estimation data are presented based on expert opinion due to the limited availability of usable data, a uniform distribution or a triangular distribution can be applied [35]. As the triangular distribution is based on a general function, its distribution can be set only by the minimum, maximum, and average points. The triangular distribution, in particular, is not significantly affected by random variables (number of data, n) [36]. Therefore, it can be said that the triangular distribution is the most suitable distribution for multi-family housing maintenance data because it is difficult to analyze and extract reliable similar cost data [37].

Therefore, this study uses the Monte Carlo simulation method, a probabilistic analysis method to enable the quantitative analysis of probabilistic maintenance costs by applying a triangular distribution. The software program @Risk is used to analyze the results according to the distributions that may occur under various scenarios produced by the Monte Carlo simulation.

The procedure and method of this study are described in Figure 1 below in detail.

First, similar studies and overlapping studies are examined and analyzed by investigating previous studies on maintenance cost analysis in relation to aged multi-family housing units. Second, in the case of South Korea, the Ministry of Land, Infrastructure, and Transport (MOLIT) presents criteria for the establishment of long-term repair plans as part of the systems and policies related to multi-family housing maintenance. As this presents reliable maintenance data, maintenance cost items can be classified and derived through the long-term repair plan practical guidelines. Third, maintenance cost cases are examined and analyzed for multi-family housing units aged 20 years or older in Seoul.

Based on the derived average maintenance costs of aged multi-family housing, a survey with experts is conducted for the application of the lower and upper limit ratios and the results are utilized as the basic data of the triangular distribution for the Monte Carlo simulation analysis. Subsequently, the derived lower and upper limit ratios are converted into costs through expert opinions based on the aged apartment house maintenance cost cases derived above, and probabilistic costs are analyzed by applying the triangular distribution of the Monte Carlo simulation analysis. Finally, the probabilistic results for maintenance costs are derived through the process above.

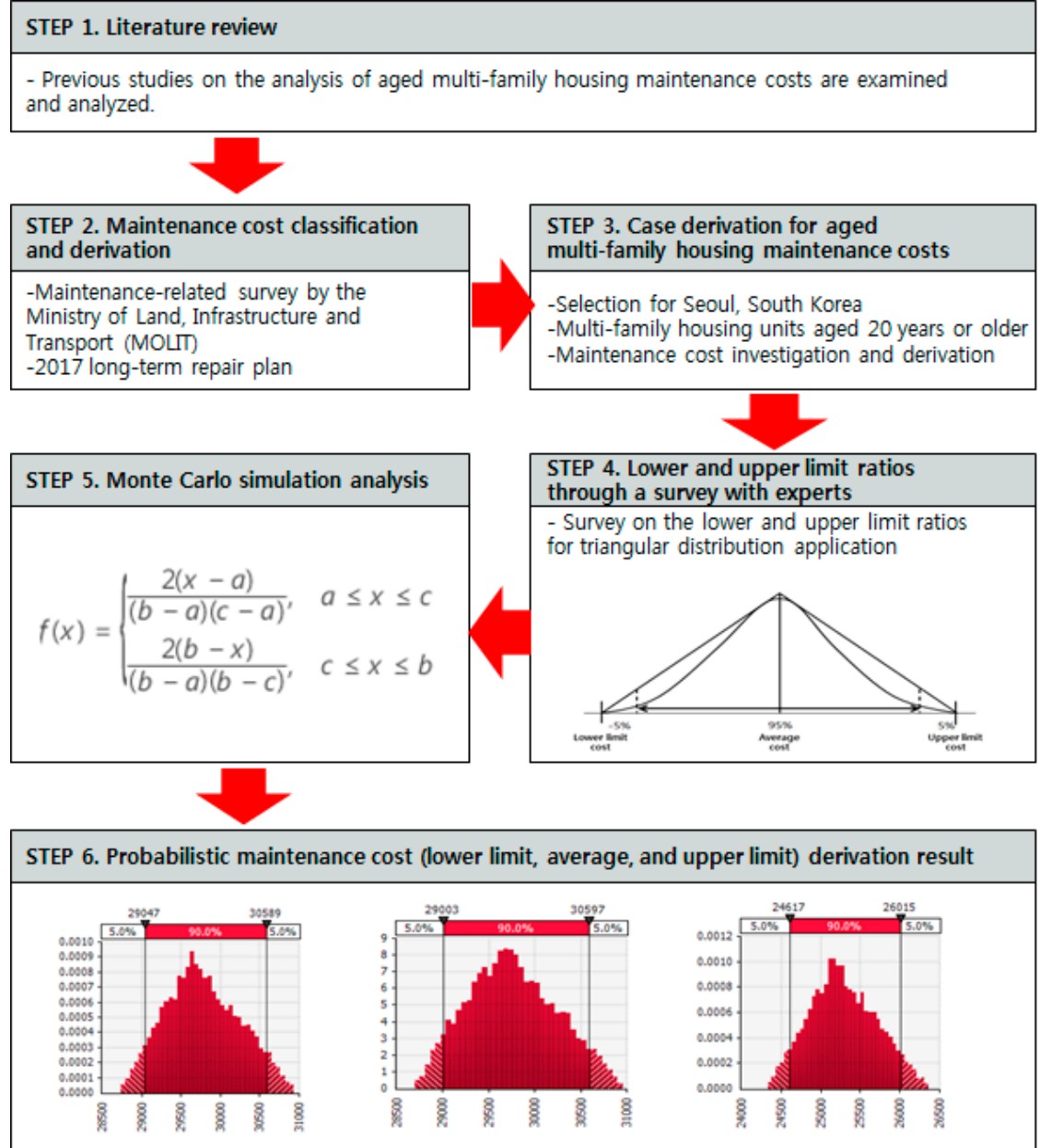

**Figure 1.** Flow of analysis methodology.

## 4. Experimental Study

### 4.1. Definition of Maintenance Cost and Itemization of Aged Multi-Family Housing Units

There are various cost and construction classification systems defined for the maintenance of aged multi-family housing units. The maintenance costs of this study are defined in terms of the repair categories, as presented by the MOLIT guidelines: Construction, electricity and information communication, machinery, and civil engineering and landscaping. Table 1 shows the detailed classification items.

**Table 1.** Classification of maintenance cost items.

| Large Category (Level 1) | Middle Category (Level 2) | Small Category (Level 3) | Applied to This Study |
|---|---|---|---|
| Construction | Building exterior | Roof | ○ |
| | | Exterior | ○ |
| | | External window/door | ○ |
| | Building interior | Ceiling | Excluding exclusive areas |
| | | Internal wall | ○ |
| | | Floor | Excluding exclusive areas |
| Electricity/information communication | Electricity/firefighting/elevator | Stairs | ○ |
| | | Spare power facility | ○ |
| | | Substation | ○ |
| | | Automatic fire detection facility | ○ |
| | | Firefighting facility | ○ |
| | | Elevator and lift | ○ |
| | | Lightning protection facility and outdoor lighting | ○ |
| | | Communication and broadcast | ○ |
| | | Boiler room and machine room | ○ |
| | | Security/crime prevention facility | ○ |
| | | Intelligent home network facility | No case application |
| Machinery | Water supply/gas/drainage/ventilation facilities | Water supply facility | ○ |
| | | Gas facility | ○ |
| | | Drainage facility | ○ |
| | | Ventilation facility | ○ |
| | Heating and hot-water supply facilities | Heating facility | ○ |
| | | Hot-water supply facility | ○ |
| Civil engineering/landscaping | Outdoor and welfare facilities | Outdoor and welfare facilities | ○ |

As presented in Table 1, ceilings and floors in the construction area were excluded from this study because some of them are included in the exclusive areas. Moreover, the intelligent home network facility in the electricity/information communication area was excluded because it is not applicable to multi-family housing projects completed over 20 years ago.

*4.2. Survey on the Maintenance Costs of Aged Multi-Family Housing*

4.2.1. Case Study Development

For the analysis of the maintenance costs of multi-family housing units aged 20 years or older, which is the purpose of this study, conditions for case selection are set as follows. First, the maintenance costs of aged multi-family housing units may vary depending on the region. In this study, maintenance costs are analyzed for the properties located in Seoul. Second, starting from 10 years after completion, multi-family housing units are often subject to maintenance issues, such as concrete neutralization, steel reinforcement corrosion, and the failure of various other facilities, and therefore experience rapidly increasing maintenance costs. Therefore, the criterion for choosing aged multi-family housing is limited to 20 years or longer after completion. Third, if the maintenance costs of aged multi-family housing units are limited to each section and process, a difficulty arises in the analysis process due to the difference in data. Therefore, the maintenance costs of this study are limited to the items in accordance with the long-term repair plan establishment criteria presented by MOLIT.

Based on the criteria above, target cases suitable for this study were selected, as shown in Table 2 below.

**Table 2.** Target case multi-family housing units.

| Region | Number of Complexes | Number of Households | Region | Number of Complexes | Number of Households |
|---|---|---|---|---|---|
| GR | 50 | 23,278 | DDM | 17 | 8934 |
| MP | 15 | 10,879 | GJ | 29 | 14,802 |
| EP | 20 | 6,721 | GC | 15 | 7,560 |
| GB | 11 | 6,007 | DJ | 30 | 20,091 |
| SD | 26 | 14,278 | JN | 31 | 21,214 |
| | Total | | | 224 | 133,764 |

In the cases included in this study, maintenance costs from 224 multi-family housing complexes (133,764 households) from 2011 to 2014 were investigated in accordance with the classification shown in Table 2. Each of the properties was aged 20 years or older and located in one of Seoul's 10 boroughs.

4.2.2. Extraction of Maintenance Costs for Aged Multi-Family Housing Units

The maintenance costs of the apartment houses aged 20 years or older in Seoul can be derived and summarized as follows in accordance with the case selection conditions above, the definition of aged multi-family housing maintenance costs, and the repair classifications set forth in the long-term repair plan establishment criteria.

First, the maintenance cost for the building exterior was found to be KRW 21,985,561,000. Second, the maintenance cost for the building interior was KRW 325,195,000. Third, the maintenance cost for the electricity/firefighting/elevator was KRW 9,854,189,000. Fourth, the maintenance cost for the water supply/gas/drainage/ventilation facilities was KRW 12,337,182,000. Fifth, the maintenance cost for the heating and hot-water supply facilities was KRW 7,517,083,000. Sixth, the maintenance cost for the outdoor and welfare facilities was KRW 6,674,856,000. Table 3 shows the average costs for each complex analyzed through the maintenance cost cases of the apartment houses aged 20 years or older in Seoul.

**Table 3.** Average maintenance costs for each complex (unit: thousand KRW).

| Category | | Total | Complex Average | % |
|---|---|---|---|---|
| Building exterior | Roof | 7,230,881 | 29,635 | 12.32% |
| | Exterior | 14,483,874 | 59,360 | 24.68% |
| | External window/door | 270,806 | 1110 | 0.46% |
| Building interior | Internal wall | 96,000 | 393 | 0.16% |
| | Stairs | 229,195 | 939 | 0.39% |
| Electricity/firefighting/elevator | Spare power facility | 80,138 | 328 | 0.14% |
| | Substation | 1,514,086 | 6205 | 2.58% |
| | Automatic fire detection facility | 141,727 | 581 | 0.24% |
| | Firefighting facility | 478,322 | 1960 | 0.81% |
| | Elevator and lift | 6,145,810 | 25,188 | 10.47% |
| | Lightning protection facility and outdoor lighting | 96,087 | 394 | 0.16% |
| | Communication and broadcast | 447,608 | 1834 | 0.76% |
| | Boiler room and machine room | - | - | 0.00% |
| | Security/crime prevention facility | 950,411 | 3895 | 1.62% |
| Water supply/gas/drainage/ventilation facilities | Water supply facility | 10,398,507 | 42,617 | 17.72% |
| | Gas facility | 17,446 | 72 | 0.03% |
| | Drainage facility | 1,920,229 | 7870 | 3.27% |
| | Ventilation facility | 1,000 | 4 | 0.00% |
| Heating and hot-water supply facilities | Heating facility | 7,230,836 | 29,635 | 12.32% |
| | Hot-water supply facility | 286,247 | 1173 | 0.49% |
| Outdoor | Outdoor and welfare facilities | 6,674,856 | 27,356 | 11.37% |
| | Total | 58,694,066 | 240,549 | 100.00% |

As shown in Table 3, the average cost for each apartment complex was KRW 240,549,000. The average cost for the building exterior was KRW 59,360,000. The average cost for the elevator and lift of the electricity/firefighting/elevator area was KRW 25,188,000. The average cost for the water supply facility of the water supply/gas/drainage/ventilation facilities area was KRW 42,617,000. Moreover, the average cost for the roof of the building exterior area was KRW 29,635,000, and the average cost for the outdoor and welfare facilities of the outdoor area was KRW 27,356,000.

### 4.3. Expert Opinions and Surveys

To present the planned maintenance of aged multi-family housing, it is necessary to investigate the inputted maintenance costs and analyze the probabilistic cost fluctuations to estimate the upper and lower limits for maintenance costs. Accurate estimation of maintenance costs is difficult through statistical analysis of limited existing data, and the addition of expert opinions may contribute to improving the accuracy of the maintenance cost analysis.

Based on the maintenance costs derived in Table 3, expert interviews and surveys were conducted to analyze probabilistic costs. Table 4 summarizes the overview of the expert interviews and surveys for the analysis of the probabilistic cost fluctuations related to aged multi-family housing.

**Table 4.** Overview of expert interviews for analyzing the lower and upper limit ratios of aged multi-family housing maintenance costs.

| Category | Detailed Contents |
|---|---|
| Survey purpose | ○ Interviews for applying the lower and upper limit ratios of aged multi-family housing cost items |
| Survey period | ○ August, 2018 (approximately 1 month) |
| Survey target | ○ Experts in research institutes related to the construction area (7)<br>○ Construction company maintenance experts (10)<br>○ University professors in the departments of architecture and civil engineering (3) |
| Survey results | ○ Interviews with 20 industry/academia/research experts with the experience of more than 10 years in multi-family housing maintenance |

In accordance with the expert interviews summarized above, the lower and upper limit ratios for each item of the maintenance costs of the aged multi-family housing facilities were surveyed with 20 experts. For this, based on the apartment house maintenance cost items presented in Table 4, the lower and upper limit ratios based on the average costs were surveyed as follows.

First, in the case of the building exterior, the lower and upper limit ratios based on the average were −3.1% and 4.5% for the roof, −2.3% and 3.5% for the exterior, and −1.5% and 2.2% for the external window/door.

Second, in the case of the building interior, the lower and upper limit ratios based on the average were −1.6% and 2.7% for the internal wall and −1.4% and 2.3% for the stairs.

Third, in the case of the electricity/firefighting/elevator, the lower and upper limit ratios based on the average were −2.7% and 4.0% for the spare power facility, −2.7% and 4.4% for the substation, −1.4% and 1.9% for the automatic fire detection facility, −1.2% and 2.0% for the firefighting facility, −3.5% and 4.7% for the elevator and lift, −1.2% and 1.5% for the lightning protection facility and outdoor lighting, −1.2% and 1.6% for communication and broadcast, −2.6% and 3.9% for the boiler room and machine room, and −1.2% and 1.7% for the security/crime prevention facility.

Fourth, in the case of the water supply/gas/drainage/ventilation facilities, the lower and upper limit ratios based on the average were −2.8% and 4.5% for the water supply facility, −1.8% and 2.6% for the gas facility, −2.8% and 3.8% for the drainage facility, and −1.3% and 1.6% for the ventilation facility.

Fifth, in the case of the heating and hot-water supply facilities, the lower and upper limit ratios based on the average were −3.3% and 4.6% for the heating facility and −2.4% and 4.2% for the hot-water supply facility.

Sixth, in the case of the outdoor and welfare facilities, the lower and upper limit ratios based on the average were found to be −3.1% and 4.4%.

As can be seen from the expert interview results above, the items with large fluctuations among the maintenance cost items are as follows. First, the item with the highest lower limit fluctuation ratio was the elevator and lift (−3.5%), followed by the heating facility (−3.3%), the roof as well as the outdoor and welfare facilities (−3.1% each), and the water supply facility (−2.8%). Second, the item with the highest upper limit fluctuation ratio was the elevator and lift (4.7%), followed by the heating facility (4.6%), the roof as well as the water supply facility (4.5% each), and the outdoor and welfare facilities (4.4%).

Therefore, probabilistic cost fluctuation analysis was conducted in this study based on the roof of the building, elevator and lift, water supply facility, heating facility, and outdoor and welfare facilities, which exhibited high lower and upper limit fluctuation ratios among the maintenance cost items above. Thus, the complex average costs presented in Table 4 were converted into lower and upper limit ratios, as shown in Table 5.

**Table 5.** Conversion of the complex average costs into the lower and upper limit ratios for each maintenance cost item.

| Category | | Complex Average Maintenance Costs (Unit: Thousand KRW) | | |
|---|---|---|---|---|
| | | Lower Limit Cost | Average Cost | Upper Limit Cost |
| Building exterior | Roof | 28,731 | 29,635 | 30,969 |
| Electricity/firefighting/elevator | Elevator and lift | 24,319 | 25,188 | 26,372 |
| Water supply/gas/drainage/ventilation facilities | Water supply facility | 41,424 | 42,617 | 44,513 |
| Heating and hot-water supply facilities | Heating facility | 28,672 | 29,635 | 30,983 |
| Outdoor | Outdoor and welfare facilities | 26,508 | 27,356 | 28,560 |

## 4.4. Probabilistic Maintenance Cost Analysis through Monte Carlo Simulation

Based on the average costs for each maintenance cost item of the multi-family housing units derived in Section 4.3, the probabilistic cost analysis was conducted by applying the Monte Carlo simulation method, a probabilistic methodology, to the lower and upper limit costs. For this, Microsoft Excel spreadsheets with the Monte Carlo simulation formula were utilized alongside Palisade's @Risk. The triangular distribution from the @Risk software was applied, and the number of simulation repetitions was set to 10,000. The random number was fixed to 1.

Based on the results obtained from Table 5, the Monte Carlo simulation analysis was conducted by applying the lower and upper limit costs based on the complex average maintenance costs using the triangular distribution as the basic distribution. The results are summarized in Table 6 and the lower and upper limits are illustrated in Table 7.

**Table 6.** Probabilistic cost analysis results for average maintenance costs.

| Category | | Complex Average Maintenance Costs (Unit: Thousand KRW) | |
|---|---|---|---|
| | | Average Cost Input Value | Probabilistic Cost Analysis Result |
| Building exterior | Roof | 29,635 | 29,780 |
| Electricity/firefighting/elevator | Elevator and lift | 25,118 | 25,291 |
| Water supply/gas/drainage/ventilation facilities | Water supply facility | 42,617 | 42,844 |
| Heating and hot-water supply facilities | Heating facility | 29,635 | 29,763 |
| Outdoor | Outdoor and welfare facilities | 27,356 | 27,482 |

**Table 7.** Lower limit, average, upper limit cost analysis results for major complex average maintenance cost items.

| Category | Analysis Graph | Probabilistic Cost Analysis Result (Unit: Thousand KRW) | |
|---|---|---|---|
| Roof |  | Lower limit cost | 28,774 |
| | | Average cost | 29,780 |
| | | Upper limit cost | 30,930 |
| Elevator and lift |  | Lower limit cost | 24,335 |
| | | Average cost | 25,291 |
| | | Upper limit cost | 26,367 |
| Water supply facility |  | Lower limit cost | 41,435 |
| | | Average cost | 42,844 |
| | | Upper limit cost | 44,476 |
| Heating facility |  | Lower limit cost | 28,695 |
| | | Average cost | 29,763 |
| | | Upper limit cost | 30,964 |
| Outdoor and welfare facilities |  | Lower limit cost | 26,518 |
| | | Average cost | 27,482 |
| | | Upper limit cost | 28,545 |

As shown in Table 7, for the roof of the building exterior, the lower limit cost changed by −3.38% to KRW 28,774, 000, and the upper limit cost changed by 3.86% to KRW 30,930,000 based on an average cost of KRW 29,780,000. For the elevator and lift, the lower limit cost changed by −3.79% to KRW 24,335,000, and the upper limit cost changed by 4.25% to KRW 26,367,000 based on an average cost of KRW 25,291,000. For the water supply facility, the lower limit cost changed by −3.29% to KRW 41,435,000, and the upper limit cost changed by 3.81% to KRW 44,476,000 based on an average cost of KRW 42,844,000. For the heating facility, the lower limit cost changed by −3.59% to KRW 28,695,000, and the upper limit cost changed by 4.03% to KRW 30,964,000 based on an average cost of KRW 29,763,000. For the outdoor and welfare facilities, the lower limit cost changed by −3.51% to KRW 26,518,000 and the upper limit cost changed by 3.87% to KRW 28,545,000 based on an average cost of KRW 27,482,000.

## 5. Discussion

The fluctuation ranges of the lower and upper limit costs were analyzed based on the average costs of the maintenance cost items defined in this study through an expert survey on the lower and upper limit ratios for each maintenance cost item. Among these items, the lower and upper limit variation rates were analyzed for specific elements of each group—roofs (building exterior), elevators and lifts (electricity/firefighting/elevator), water supply (water supply/gas/drainage/ventilation facilities), heating (heating and hot-water supply facilities), and outdoor units and welfare facilities—for the following reasons.

The cost of roof repairs is considered to be an important management factor among maintenance cost items because of the obsolescence of materials and facilities due to the building's exposure to the exterior environment. In the case of elevators and lifts, the cost of frequent repairs and replacements due to the rapid increase in fatigue is also considered to be an important management factor among maintenance cost items. For the water supply facility, the occurrence of pipe repairs after the completion of building construction is low, but there are more significant costs related to the eventual replacement of pipes that are required for continuous use in apartments older than 20 years. With respect to heating and hot-water supply facilities, significant costs are incurred on the replacement of facilities' piping, such as heat exchangers, heat piping, and heating pumps. Finally, outdoor units and welfare facilities are considered to be important management factors among the maintenance cost items, as many elements, such as children's playgrounds, roads and sidewalk blocks, and landscaping, incur frequent repair and replacement costs due to increased fatigue. As noted above, it is not easy to predict maintenance costs because the facilities and infrastructure of multi-family apartment buildings are replaced and renovated in accordance with the way residents live and the unexpected failure of various facilities.

In the construction sector, most studies use general data statistical analysis methods to predict maintenance costs, but they are somewhat limited to the extraction and generalization of the results into representative values. Making up for such limitations of these methods, this study derives representative analyses from Monte Carlo simulations using expert population surveys to generalize empiricism and estimate maintenance costs. Therefore, if the proposed Monte Carlo simulation method is utilized, the variability in maintenance costs due to lower and upper limit ratios can be analyzed and estimated by applying the knowledge of experts. This appears to constrain the sampling error to a minimum. Moreover, the results of the method proposed in this study can be utilized as reference data to predict and analyze the variability in aged multi-family housing maintenance costs for sustainable construction. It is believed that the maintenance cost analysis method presented herein could also be applied throughout Korea and other geographical environments at similar latitudes and longitudes.

## 6. Conclusions

The present study was conducted to provide useful information on multi-family housing maintenance by investigating and analyzing the maintenance costs of aged properties for sustainable construction. For this purpose, the maintenance costs of multi-family housing units aged 20 years or older in Seoul were investigated and analyzed. Moreover, the probabilistic cost analysis was conducted using the Monte Carlo simulation method. The results of this study conducted through the process above are summarized as follows.

First, in this study, the maintenance cost items were derived by referring to the 2017 long-term repair plan practical guidelines of the MOLIT. The construction area was classified into the building exterior (roof, exterior, and external window/door) and the building interior (internal wall and stairs). The electricity/information communication area was classified into electricity/firefighting/elevator (spare power facility, substation, automatic fire detection facility, firefighting facility, elevator and lift, lightning protection facility and outdoor lighting, communication and broadcast, boiler room and machine room, and security/crime prevention facility). The machinery area was classified into water supply/gas/drainage/ventilation facilities (water supply facility, gas facility, drainage facility, and ventilation facility) and heating and hot-water supply facilities (heating facility and hot-water supply facility). The civil engineering/landscaping area was classified into outdoor and welfare facilities. Therefore, a total of 21 detailed maintenance cost items were derived. In addition, the maintenance costs of a total of 224 apartment complexes (133,764 households) aged 20 years or older in the 10 boroughs of Seoul were investigated as follows. The building exterior and the building interior represented 37.46% and 0.55% of the total maintenance cost. The electricity/firefighting/elevator represented 16.79%, the water supply/gas/drainage/ventilation facilities represented 21.02%, and the heating and hot-water supply facilities represented 12.81%. In addition, the outdoor and welfare

facilities represented 11.37%. As for the detailed items, the exterior of the building exterior exhibited the highest maintenance cost (24.68%), followed by the water supply facility (17.72%), the roof of the building exterior (12.32%), the heating facility of the heating and hot-water supply facilities (12.32%), and the outdoor and welfare facilities (11.37%).

Second, the triangular distribution was applied based on the maintenance costs above and the lower and upper limit ratios were analyzed based on the average costs of the maintenance cost items through expert interviews. The results were analyzed using @Risk. As a result, for the roof of the building exterior, the lower limit cost changed by −3.38% and the upper limit cost changed by 3.86% based on the average cost. For the elevator and lift of the electricity/firefighting/elevator, the lower limit cost changed by −3.79% and the upper limit cost changed by 4.25% based on the average cost. For the water supply facility of the water supply/gas/drainage/ventilation facilities, the lower limit cost changed by −3.29% and the upper limit cost changed by 3.81% based on the average cost. For the heating facility of the heating and hot-water supply facilities, the lower limit cost changed by −3.59% and the upper limit cost changed by 4.03% based on the average cost. For the outdoor and welfare facilities, the lower limit cost changed by −3.51% and the upper limit cost changed by 3.87% based on the average cost. Therefore, the variability in the maintenance costs according to the probability can be estimated using the proposed Monte Carlo simulation method.

This study was conducted to provide useful information for the analysis of maintenance costs to improve planning for maintenance costs and repairs for aged multi-family housing units in Seoul. However, we acknowledge that generalization of the results of the maintenance cost analysis presented in this study is limited due to the fact that more maintenance data could not be secured. In addition, the cases were limited to one geographic area. Moreover, the probabilistic maintenance cost fluctuation range had to be analyzed by applying risk factors to the maintenance of the properties, but they could not be applied. Therefore, in future studies, we will actively seek to broaden the scope of the sample group, not only in terms of numbers, but also to take into account the diversity and reflectivity of the study to overcome the limitations of the current work.

**Author Contributions:** Conceptualization, M.P. and Y.A.; Investigation, N.K. and S.L.; Methodology, M.P., J.L., S.L. and Y.A.; Supervision, J.L. and Y.A.; Validation, M.P. and N.K.; Writing—original draft, M.P. and Y.A.; Writing—review & editing, M.P., N.K., J.L. and Y.A.

**Acknowledgments:** This work was supported by the Korea Institute of Energy Technology Evaluation and Planning (KETEP) and the Ministry of Trade, Industry & Energy (MOTIE) of the Republic of Korea (No. 20172010000370).

**Conflicts of Interest:** The authors declare no conflict of interest.

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
