# Peer review of "Probabilistic Maintenance Cost Analysis for Aged Multi-Family Housing"

_sustainability, doi:10.3390/su11071843_

Round 1

Reviewer 1 Report

The paper addresses a fairly interesting topic. The methodology is appropriate. The bibliography is wide and adequate.

The paper is well structured but in some parts it is repetitive. The following repetitions should be deleted:

- lines 47-48 and lines 70-71;

- lines 102-104, lines 122-123 and lines 171-173;

- lines 157-160 and lines 177-180;

- lines 234-236 and lines 310-312.

Section 5 "Discussion" is not much detailed and partly repeats the data and results already presented in section 4.4 "Probabilistic Maintenance Cost Analysis through Monte Carlo Simulation".

Finally, the paper shows a considerable limitation of the research. As the authors themselves state, the study was conducted on a restricted geographic area and the overall application of the results of the analysis of maintenance costs is limited; therefore, further research is required in the future to understand if the probabilistic maintenance costs fluctuation range obtained for the case study is similar also in other geographical areas and how the results of this research can be extended to other contexts.

Author Response

We wish to re-submit the attached manuscript as an Original Article. The manuscript ID is 445106.

The manuscript has been rechecked and appropriate changes have been made in accordance with the reviewers’ suggestions. All of the comments raised by the reviewers are addressed. The responses to their comments have been prepared and are given below. The changes made in the manuscript are also included.

We thank the editor and the reviewers for your thoughtful suggestions and insights, which have enriched the manuscript and produced a better and more balanced account of the research. We hope that the revised manuscript is now suitable for publication in your journal.

Reviewer 2 Report

There is a need to close the loop in the research methodology and discussion of findings to explain the unique value of applying probabilistic analysis to maintenance costs or tweak the title to fit the main message in these sections. 

The minimum sample size is 30. The sample size is too small for any generalizations to be made.

Author Response

(The authors gave the same response as above.)

Round 2

Reviewer 2 Report

The authors have addressed the feedback from the first review satisfactority.